# Guided Search Strategies in Non-Serializable Environments with Applications to Software Engineering Agents

**Karina Zainullina** [* 1]  **Alexander Golubev** [* 1]  **Maria Trofimova** [* 1]  **Sergei Polezhaev** [1]  **Ibragim Badertdinov** [1]
**Daria Litvintseva** [1]  **Simon Karasik** [1]  **Filipp Fisin** [1]  **Sergei Skvortsov** [1]  **Maksim Nekrashevich** [1]  **Anton Shevtsov** [1]
**Boris Yangel** [1]

## Abstract

Large language models (LLMs) have recently achieved remarkable results in complex multi-step tasks, such as mathematical reasoning and agentic software engineering. However, they often struggle to maintain consistent performance across multiple solution attempts. One effective approach to narrow the gap between average-case and best-case performance is guided test-time search, which explores multiple solution paths to identify the most promising one. Unfortunately, effective search techniques (e.g. MCTS) are often unsuitable for *non-serializable* RL environments, such as Docker containers, where intermediate environment states cannot be easily saved and restored. We investigate two complementary search strategies applicable to such environments: 1-step lookahead and trajectory selection, both guided by a learned action-value function estimator. On the SWE-bench Verified benchmark, a key testbed for agentic software engineering, we find these methods to double the average success rate of a fine-tuned Qwen-72B model, achieving $40.8\%$, the new state-of-the-art for open-weights models. Additionally, we show that these techniques are transferable to more advanced closed models, yielding similar improvements with GPT-4o.

## 1. Introduction

The rise of large language models (LLMs) has driven significant advancements across multiple domains. However, when it comes to tasks requiring heavy reasoning or agentic capabilities, a major challenge persists: while the models occasionally achieve exceptional results, their average per-

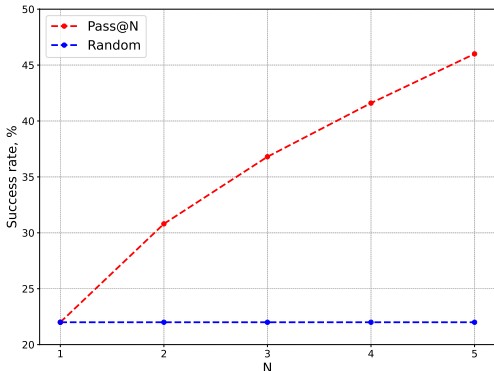

*Figure 1.* The comparison of two evaluation protocols for a GPT-4o-based agent: **Pass@$N$** and **random sampling**. The x-axis shows the number of attempts, the y-axis shows the average success rate (i.e. the percentage of correct solutions).

formance often falls short of their demonstrated potential.

To illustrate this challenge, Figure 1 depicts the performance of a reasonably capable GPT-4o-based agent (Yang et al., 2024) on SWE-bench Verified (Jimenez et al., 2024) under two evaluation protocols:

- **Pass@$N$**: the agent makes $N$ attempts to solve each problem instance, and a problem is considered solved if at least one attempt succeeds. This protocol demonstrates the model's potential capability ceiling.

- **Random sampling**: a problem is considered solved if a randomly selected solution from $N$ attempts is correct.

This example highlights that LLMs exhibit high variance in success rate across different attempts. It is common for sequential tasks where multiple correct decisions must be chained together: even if a model makes $80\%$ of individual decisions correctly, the probability of sampling a complete successful trajectory can decrease exponentially with the

---

*Equal contribution  [1]Nebius. Correspondence to: Boris Yangel <byangel@nebius.com>.

*Proceedings of the $42^{nd}$ International Conference on Machine Learning*, Vancouver, Canada. PMLR 267, 2025. Copyright 2025 by the author(s).

number of required steps (i.e. $0.8^T$ for $T$ steps), depending on the ability of the model to self-correct.

In domains that involve complex reasoning, such as mathematical problem solving, recent work has demonstrated improvements in performance consistency through two main approaches. Outcome Reward Models (ORMs) (Cobbe et al., 2021; Mudgal et al., 2024) predict the correctness of the final answer that is then used for solution reranking. Process Reward Models (PRMs) (Lightman et al., 2024; Wang et al., 2024a; Uesato et al., 2022) evaluate the correctness of intermediate steps. Step-level evaluations provided by PRMs can be combined into a single score for reranking, but they also enable classical search methods such as Beam Search (Setlur et al., 2024), Best-First Search (BeFS) (Koh et al., 2024), Depth-First Search (DFS) (Yao et al., 2023a), and Monte-Carlo Tree Search (MCTS) (Kocsis & Szepesvári, 2006; Gao et al., 2024; Xie et al., 2024; Hao et al., 2023; Putta et al., 2024). Search methods improve consistency by systematically exploring the space of solutions instead of relying on chance.

Unfortunately, many agentic systems operate in *non-serializable environments* (formally defined in Section 2.3), where intermediate states cannot be saved, replicated, or reversed. Non-serializability prevents the use of search methods that require multiple roll-outs from the same state, significantly limiting the available forms of exploration. For example, Monte-Carlo Tree Search (MCTS) becomes impossible in these settings as it relies on the ability to re-explore previously visited states.

Our main contributions are as follows:

- We introduce the notion of *non-serializable* RL environments, with Docker containers being one important example, and highlight the limits to the applicability of powerful guided search techniques such as MCTS to these environments.

- We identify and systematically study two guided search methods applicable to non-serializable environments, using SWE-agent (Yang et al., 2024) on SWE-bench Verified (Jimenez et al., 2024) as a testbed. Our findings reveal that these techniques and their combination can significantly bridge the gap between peak and average agent capabilities even under non-serializability constraints. Moreover, we observe favourable performance scaling with more test-time computation.

- We apply the proposed techniques on top of a state-of-the-art open-weights LLM and demonstrate the best-in-class success rate of $40.8\%$ on the SWE-bench Verified.

- We demonstrate that the proposed method is equally applicable to powerful proprietary models.

## 2. Preliminaries, Definitions, and Notation

### 2.1. Problem Setup and Notation

Following (Murphy, 2024) we formalize our setting as a Partially Observable Markov Decision Process (POMDP) with discounted rewards, defined by the tuple $\langle \mathcal{Z}, \mathcal{A}, \Omega, W, \mathcal{O}, \mathcal{R}, \gamma \rangle$, where

- $\mathcal{Z}$ is the set of environment states,

- $\mathcal{A}$ is the set of actions,

- $\Omega$ is the set of observations,

- $W, \mathcal{O}, \mathcal{R}$ are stochastic transition, stochastic observation, and reward functions,

- $\gamma \in [0, 1)$ is the discount factor.

At each time step $t$, the environment is in state $z_t \in \mathcal{Z}$, which is not directly observable by the agent. Instead, the agent receives observation $o_t \sim \mathcal{O}(o \mid z_t)$ and maintains its internal state $s_t$, defined as the history of past observations and actions $s_t = (o_0, a_0, o_1, a_1, \ldots, o_t)$, which we also refer to as *trajectory*. Based on $s_t$, the agent follows policy $\pi$ to issue an action $a_t \sim \pi(a \mid s_t) \in \mathcal{A}$. The environment then transitions to $z_{t+1} \sim W(z \mid z_t, a_t)$ and provides reward $r_t = \mathcal{R}(z_t, a_t, z_{t+1})$.

We consider an episodic setting where each episode terminates upon reaching one of the terminal states $z_T \in \mathcal{Z}_{\text{terminal}} \subset \mathcal{Z}$. We refer to trajectories that reach a terminal state as *complete* trajectories, denoted as $s_T$. In our setting, we focus on sparse reward environments, where non-zero rewards are only provided at terminal states, i.e. $r_t = 0$ for $t < T - 1$ and $r_{T-1} \in \{0, 1\}$. This reward structure naturally arises in scenarios where ground truth is provided by a verifier that evaluates the entire trajectory upon episode completion. For any step $t$, the return-to-go is defined as the discounted terminal reward $R_t \triangleq \gamma^{T-t-1} r_{T-1}$.

### 2.2. Guided Search

Given a problem encoded using the POMDP formulation described above, we formalize guided search methods through the notion of inference operators, rules that induce the distribution over trajectories given an initial state. In its most general form, an inference operator is a distribution $\mathcal{I}(s_T \mid s_0)$. The most straightforward inference operator is the *base* inference operator $\mathcal{I}[\pi](s_T \mid s_0)$ that uses some policy $\pi$ to generate complete trajectories according to the following

procedure:

$$a_t \sim \pi(a \mid s_t),$$
$$z_{t+1} \sim W(z \mid z_t, a_t),$$
$$o_{t+1} \sim \mathcal{O}(o \mid z_{t+1}), \quad (1)$$
$$s_{t+1} \triangleq (\underbrace{o_0, a_0, \ldots, o_t}_{s_t}, a_t, o_{t+1})$$

until a terminal state $z_{t+1} = z_T \in \mathcal{Z}_{\text{terminal}}$ is reached.

We can build other forms of *inference operators* $\mathcal{I}$ with the aim to improve upon $\mathcal{I}[\pi](s_T \mid s_0)$ in terms of expected reward, where the expectation is taken over the stochasticity of the operator itself and the starting distribution of problems $P$:

$$\mathbb{E}_{s_0 \sim P, \ s_T \sim \mathcal{I}(s|s_0)}[r_{T-1}] \to \max. \quad (2)$$

Improvement can be achieved by leveraging access to the base policy $\pi$ and its statistics such as the action-value function $Q_\pi(s, a)$. We call inference operators that leverage action-value functions *guided search* operators and denote them as $\mathcal{I}[\pi, Q_\pi](s_T \mid s_0)$. One trivial example of a guided search inference operator is an operator induced by some policy improvement operator $\mathcal{PI}[\pi, Q_\pi]$ (Sutton & Barto, 1998):

$$\mathcal{I}[\pi, Q_\pi](s \mid s_0) \triangleq \mathcal{I}\big[\mathcal{PI}[\pi, Q_\pi]\big](s \mid s_0). \quad (3)$$

However, as we show later, more general forms of guided search are possible.

### 2.3. Non-Serializable Environment

**Definition 2.1.** A *non-serializable environment* in the context of reinforcement learning (RL) is an environment, which states $z_t \in \mathcal{Z}$ cannot be serialized and de-serialized at an arbitrary time step $t$, and acting in a state changes the state in-place, potentially irreversibly.

Non-serializability has the following implications:

- **No rollbacks**: once action $a_t$ is taken in state $z_t$ to obtain $z_{t+1} = W(z_t, a_t)$, there is no procedure to revert the environment state back to $z_t$.

- **No state copying**: we cannot create multiple instances of $z_t$ to obtain observations resulting from taking different actions in this state.

- **No branching**: methods requiring multiple unrolls from the same state (e.g. MCTS) cannot be directly applied.

- **Limited forms of exploration**: agent must proceed strictly forward through the transition function $W$.

One example of such environment is SWE-agent (Yang et al., 2024), a framework for solving GitHub issues through interactions with a Docker container. While Docker supports container checkpointing through CRIU (Checkpoint/Restore In Userspace) (CRIU community, 2019), several technical limitations prevent reliable state serialization (Andrijauskas et al., 2024; Dash, 2022). Specifically, CRIU cannot guarantee consistent preservation of shared memory segments and inter-process communication channels. During its operation, SWE-agent might spawn complex background process hierarchies. When restored, these processes face issues with PID reassignment and broken parent-child relationships, disrupting the normal execution flow.

An alternative approach to replicate an environment state $z_t$ is to replay the action sequence $(a_0, \ldots, a_{t-1})$ in a freshly initialized Docker container. However, this approach is computationally expensive as it requires to re-run all actions from the beginning of trajectory, potentially including time-consuming ones such as compilation or running tests. Moreover, the transition function $W(z, a)$ exhibits stochastic behavior due to multiple sources of non-determinism: process scheduling, inter-process communications and access to entropy sources like current time or external systems. Consequently, executing the same action sequence multiple times may result in different state trajectories, making it impossible to reliably reproduce a specific state $z_t$ through action replay.

Overall, the lack of serialization ability mirrors real-world software engineering scenarios, where changes accumulate sequentially and mistakes must be mitigated within the forward flow of the solution process.

## 3. Guided Search in Non-serializable Environments

In this section, we describe several guided search techniques applicable to non-serializable environments. We assume that we are given a base policy $\pi(a \mid s)$ and its action-value function $Q_\pi(s, a)$. In Section 4 we explore different ways to approximate $Q_\pi(s, a)$ with a learned model.

### 3.1. 1-step Lookahead

As discussed above, one way to build a guided search inference operator is by inducing it via a policy improvement operator. A well-known policy improvement operator applicable to non-serializable environments is 1-step lookahead (Sutton & Barto, 1998):

$$\mathcal{PI}[\pi, Q_\pi](a \mid s) = \mathbb{1}\left[a = \arg\max_{a' \in \mathcal{A}} Q_\pi(s, a')\right]. \quad (4)$$

When the action space is large, as it is the case for LLM-based agents, we can utilize its sample-based version (Hu-

---

**Algorithm 1** Sample-based 1-step lookahead

---

1: **Input:** base policy $\pi$, number of action candidates $K$, critic model $Q_\pi$, environment $E$
2: Initialize $s \leftarrow E.\text{init}()$
3: **repeat**
4:    **Initialize** listOfActions $\leftarrow$ []
5:    **Initialize** listOfQValues $\leftarrow$ []
6:    **for** $k = 1$ **to** $K$ **do**
7:      $a' \leftarrow$ sample from $\pi(a \mid s)$
8:      listOfQValues.append($Q_\pi(s, a')$)
9:      listOfActions.append($a'$)
10:    **end for**
11:    $a^* \leftarrow$ listOfActions[$\arg\max$(listOfQValues)]
12:    $s \leftarrow E.\text{step}(s, a^*)$
13: **until** $s$ is terminal
14: **return** $s$

---

bert et al., 2021):

$$\mathcal{PI}[\pi, Q_\pi](a \mid s) =$$
$$= \mathbb{1}\left[a = \arg\max_{a' \in \{a'_1, a'_2, \ldots, a'_K\}} Q_\pi(s, a')\right], \quad (5)$$

where $a'_k \sim \pi(a \mid s)$.

The pseudocode for the sample-based version of this search procedure is presented in Algorithm 1. Notice how this search strategy eliminates the need to observe outcomes of all sampled actions $a'_i \sim \pi(a \mid s)$. Instead, it estimates $Q_\pi(s, a)$ for each candidate action and proceeds with the one that has the highest action-value, guiding the trajectory towards more promising paths reachable by the base policy $\pi$. Some additional benefits of this strategy are:

- It is straightforward to implement this strategy on top of any existing base policy and combine it with other guided search methods.

- If the base policy is an LLM, this strategy has minimal impact on inference costs: it only increases the number of generated output tokens (by a factor of $K$, the number of action candidates), while the primary factor that influences agentic inference costs is the number of input tokens, especially in scenarios involving long trajectories.

- Action candidates can be generated in parallel, reducing impact on inference latency. However, one would still have to wait for the generation of the longest action candidate, and then for the estimation of its action-value.

### 3.2. Trajectory Selection

Another way to build a guided search operator is to leverage the action-value function to select the most promising candi-

---

**Algorithm 2** Trajectory selection

---

1: **Input:** base policy $\pi$, number of runs $N$, critic model $Q_\pi$, environment $E$
2: **Initialize** listOfTrajectories $\leftarrow$ []
3: **Initialize** listOfQValues $\leftarrow$ []
4: **for** $i = 1$ **to** $N$ **do**
5:    $s_T \leftarrow$ unrollPolicy($\pi$, $E$)
6:    $(*s_{T-1}, a_{T-1}, o_T) \leftarrow s_T$
7:    listOfTrajectories.append($s_T$)
8:    listOfQValues.append($Q_\pi(s_{T-1}, a_{T-1})$)
9: **end for**
10: $s^* \leftarrow$ listOfTrajectories[$\arg\max$(listOfQValues)]
11: **return** $s^*$

---

date from a batch of complete trajectories. Specifically, for an *almost complete* trajectory $s_{T-1} = (o_0, a_0, \ldots, o_{T-1})$, the action-value of its terminating action $a_{T-1}$ simplifies to the trajectory reward $Q_\pi(s_{T-1}, a_{T-1}) = r_{T-1}$.

Thus, we can build an inference operator implementing the trajectory selection process in the following way:

$$\mathcal{I}[\pi, Q_\pi](s \mid s_0) =$$
$$= \mathbb{1}\left[s = \arg\max_{s_T \in \{s_T^1, \ldots, s_T^N\}} Q(s_{T-1}, a_{T-1})\right], \quad (6)$$

where

$$s_T \triangleq (\underbrace{o_0, a_0, \ldots, o_{T-1}}_{s_{T-1}}, a_{T-1}, o_T),$$
$$s_T^i \sim \mathcal{I}[\pi](s \mid s_0).$$

This is a generalization of ORMs for arbitrary environments with sparse reward. The pseudocode for this procedure is presented in Algorithm 2. It directly attempts to close the gap between the average-case and best-case performance discussed in Section 1. Essentially, it approximates the pass@$N$ selection process by replacing the oracle that has access to evaluation results with a learned action-value estimator. This strategy offers several practical benefits:

- It is agnostic to the base policy, allowing easy integration with guided search based on policy improvement. Specifically, as we show in Subsection 4.2, this method can be efficiently combined with 1-step lookahead.

- Multiple trajectories can be generated in parallel, meaning that although trajectory selection requires additional computation, it has minimal impact on inference latency.

*Table 1.* The effects of guided search, Verified-50.

| Inference operator | Default | | Until Submitted | |
|---|---|---|---|---|
| | SR (%) | SEM | SR (%) | SEM |
| Qwen-based policy | 16.2 | ±1.08 | 22.8 | ±1.05 |
| Qwen-based policy + 1-step lookahead | 26.8 | ±1.08 | 32.4 | ±0.89 |
| Qwen-based policy + trajectory selection ($N = 5$) | 27.2 | ±0.1 | 32.27 | ±0.06 |
| Qwen-based policy + trajectory selection ($N = 10$) | 31.27 | ±0.12 | 37.6 | ±0.04 |
| Qwen-based policy + 1-step lookahead + trajectory selection ($N = 5$) | 36.5 | ±0.11 | 39.95 | ±0.06 |
| Qwen-based policy + 1-step lookahead + trajectory selection ($N = 10$) | **41.7** | ±0.13 | **44.07** | ±0.05 |
| GPT-4o policy | 22.0 | ±1.54 | 22.0 | ±1.54 |
| GPT-4o policy + 1-step lookahead | 27.2 | ± 1.20 | 29.2 | ±1.62 |
| GPT-4o policy + trajectory selection ($N = 5$) | 34.0 | – | 34.0 | – |
| GPT-4o policy + 1-step lookahead + trajectory selection ($N = 5$) | **40.0** | – | **40.0** | – |

*Table 2.* Comparison with other systems based on open-weights models, SWE-bench Verified.

| Method | SR (%) |
|---|---|
| Qwen-based policy + 1-step lookahead ($K = 8$) + trajectory selection ($N = 15$) | 40.8 |
| SWE-Gym (Qwen-based policy + trajectory selection ($N = 32$)) (Pan et al., 2024) | 32.0 |
| SWE-Fixer (Xie et al., 2025) | 30.2 |
| Lingma Agent + Lingma-SWE-GPT-72B (Ma et al., 2024) | 25.0 |

## 4. Experiments and Results

### 4.1. Experimental Setup

We utilize the SWE-agent scaffolding (Yang et al., 2024) to evaluate guided search techniques described in Section 3 on the SWE-bench Verified dataset (Jimenez et al., 2024). Our primary measure of interest is the **Success Rate (SR)**, defined as the percentage of problems successfully solved by the agent.

The SWE-agent scaffolding is a non-serializable environment, as we argue in Section 2.3. Moreover, we choose SWE-bench because it remains a challenging benchmark of practical interest, pushing LLM's abilities to effectively navigate and resolve complex software engineering scenarios to their limits. Test-time search strategies can further assist in addressing these challenges by enabling more efficient exploration within the constraints of non-serializable environments.

**Evaluation Methodology** SWE-agent's problem-solving process terminates under two conditions: when the agent issues a "submit" command indicating the problem is considered solved, or upon encountering an unrecoverable error (e.g. the LLM runs out of context). In the latter case, the accumulated changes in the workspace are treated as the generated patch. However, we observe that context exhaustion frequently correlates with unrecoverable mistakes mid-trajectory, leading to incorrect solutions. To enhance reliability, we implement an iterative execution strategy: the agent runs repeatedly until it terminates by issuing a "submit" command or reaches a maximum of 10 additional attempts. We refer to this approach, which can be viewed as a simple form of search, as *until submitted* regime, and use it by default in all experiments, if not stated otherwise. As shown in Table 1 and Appendix E, the approach significantly boosts the performance of a single run, enabling a better view into the effects of the techniques studied in this paper.

To better understand whether the observed performance differences are statistically significant, we run each experiment 10 times with distinct random seeds. We then compute the mean success rate and report it along with the standard error of the mean (SEM). This is particularly important when evaluating less capable agent policies, which exhibit higher performance variance due to inconsistent error-recovery behavior.

To balance comprehensive evaluation against computational constraints, we created *Verified*-50, a dataset of 50 randomly selected problems from SWE-bench Verified. This dataset allows us to compute an unbiased estimate of the success rate on the full SWE-bench Verified together with the estimation error at the cost of a single run on the full evaluation set. For the most promising setups, we additionally perform evaluation on the full evaluation set. The list of issues included in Verified-50 can be found in Appendix D.

**Training Details**   We run the majority of experiments with the LLM-based policy trained in-house. To estimate the action-value function of a policy, we train separate LLM-based models that we refer to as *critics*.

We start by adopting the SWE-bench issue collection methodology to compile an extended set of training issues (Badertdinov et al., 2024). Our training issue set consists of 6,500 issue–pull request pairs from 2,500 Python repositories, carefully filtered to avoid leakage of the SWE-bench test set, and supplemented with the SWE-bench development set.

We then apply a bootstrapping process to collect agent trajectories that we use to train the models, while simultaneously training a capable policy. The bootstrapping process consists of repeating the following steps:

1. Generate complete trajectories for all training issues using the current policy.

2. Evaluate these trajectories and incorporate them into the dataset with a terminal reward of 1 (if the issue is solved) or 0 (otherwise), annotating each trajectory with the corresponding policy version.

3. Update the current policy by fine-tuning on a curated subset of successful trajectories collected so far.

4. If not done, go to step 1.

We use Qwen2.5-72B (Bai et al., 2023) both as the initial collection policy and as the starting point for every fine-tuning iteration. The policy produced by fine-tuning on data produced by the final bootstrapping iteration, which we refer to as *Qwen-based* policy, is then used in the guided search experiments described below.

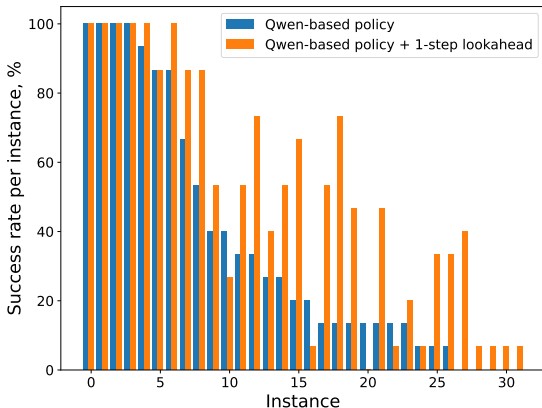

*Figure 2.* Per-instance success rate of Qwen-based policy computed over 15 runs.

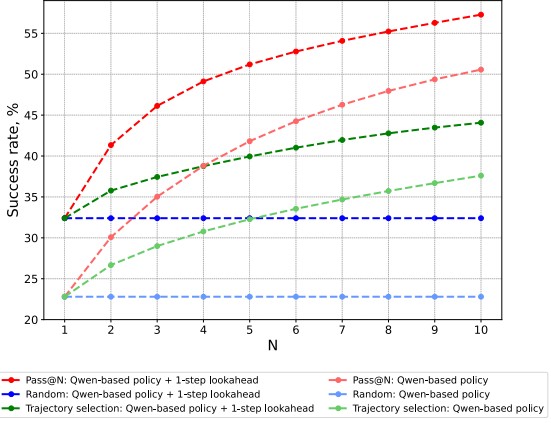

*Figure 3.* The dependency between success rate and the number of candidates $N$ in trajectory selection.

This bootstrapping process produced a total of 80,000 positive and negative trajectories. We use these trajectories to train critic models. Since our dataset contains trajectories produced by multiple policies, we incorporate policy identifiers into the critic's system prompt to condition the predicted action-value on the policy being evaluated. When running action-value prediction for policies not represented in the training data (e.g. GPT-4o), we simply pass a previously unseen policy identifier ("gpt-4o") to the critic. We use LLaMA3.1-70B (Dubey et al., 2024) to initialize critic training. Additional details on policy and critic training are provided in Appendices B, F and G.

### 4.2. Main Results

We evaluate both search strategies — 1-step lookahead and trajectory selection — as well as their superposition. The

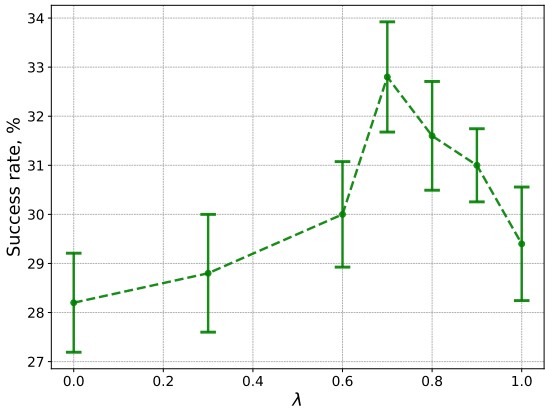

*Figure 4.* The dependency between 1-step lookahead SR and $\lambda$.

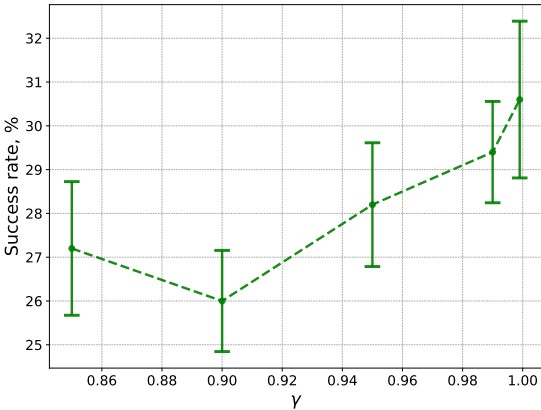

*Figure 5.* The dependency between success rate and $\gamma$ for MC.

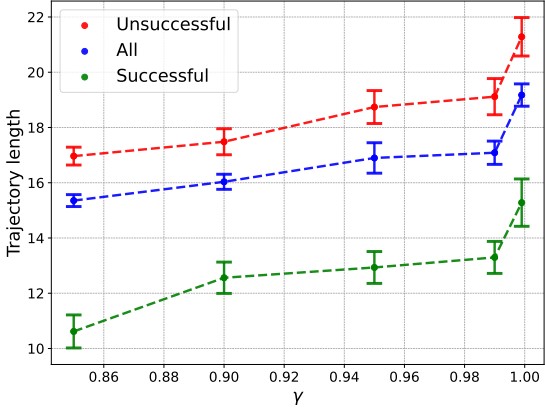

*Figure 6.* The dependency between trajectory length and $\gamma$ for MC.

evaluation uses a critic model trained with the best parameters identified in Subsection 4.3, i.e. TD(0.7), for both 1-step lookahead and trajectory selection. For 1-step lookahead, we utilize the optimal search parameters identified in Subsection 4.4: policy sampling temperature $T = 0.9$ and number of candidates $K = 4$. Trajectory selection results are presented for both $N = 5$ and $N = 10$ runs when using the Qwen-based policy. For GPT-4o, trajectory selection evaluation is limited to $N = 5$ runs without reporting SEM to reduce experimentation costs.

As shown in Table 1, both guided search methods, when used independently, achieve significant improvements over the base policy. When combined, they deliver a 2-fold improvement in success rate for both GPT-4o and Qwen-based policies. Importantly, these patterns hold consistently across both evaluation regimes: the relative ranking and magnitude of improvements remain stable whether allowing multiple attempts (Until Submitted) or terminating on first completion (Default). This consistency suggests that until submitted regime primarily addresses execution-level failures (such as context exhaustion) that affect all methods equally, rather than changing the core algorithmic dynamics that distinguish our guided search approaches.

Performance improvement on top of GPT-4o is especially notable given major challenges of this setup: a relatively modest critic's base model capabilities compared to the policy, and the absence of GPT-4o-specific trajectories in the critic's training data.

Additionally, we evaluate the effect of 1-step lookahead on per-instance success rate, presented in Figure 2 for the Qwen-based policy. Our analysis reveals that 1-step lookahead generally increases the probability of solving a problem, therefore boosting reliability. It can also boost very low success probabilities, allowing to start solving some instances where the base policy was consistently failing. GPT-4o policy demonstrates similar trends with the corresponding figure found in Appendix C.

For trajectory selection, we also investigate the relationship between the success rate and the number of candidates, $N$, when applied to the base policy and when combined with 1-step lookahead. As can be seen in Figure 3, as $N$ increases, the success rate continues to rise without reaching a visible plateau in the investigated range. Interestingly, as $N$ grows, the gap between the base policy and 1-step lookahead narrows for both pass@$N$ and trajectory selection setups, suggesting that both search methods might eventually converge to the same solutions provided enough compute. In practice, one should carefully select the appropriate values for $K$ and $N$ to achieve the desired trade-off between computational costs, latency, and performance.

We also evaluate the Qwen-based policy with 1-step looka-

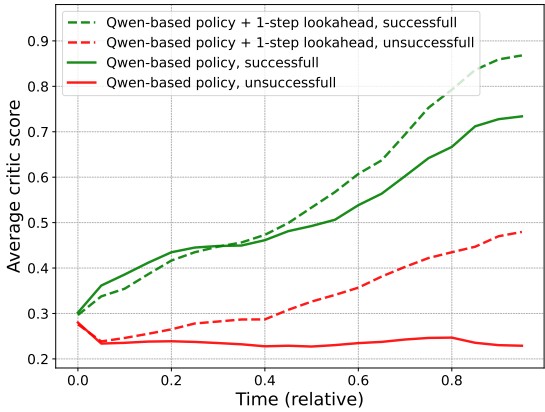

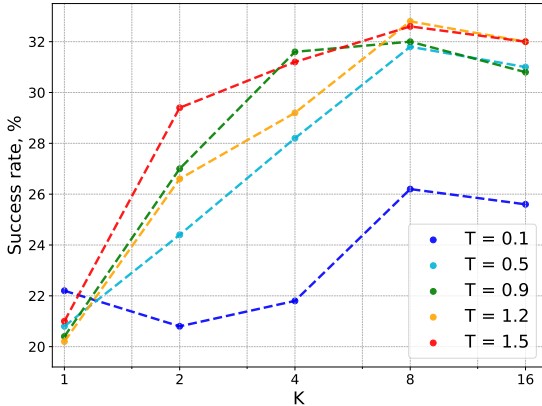

*Figure 7.* Critic learns to distinguish successfull and unsucessfull trajectories produced by Qwen-based policy.

*Figure 8.* Success rate of a TD(0.8) critic with varying $K$ and $T$.

head and trajectory selection with $N = 15$, $T = 0.9$, and $K = 8$ on the full SWE-bench Verified set, where it achieves **40.8**% success rate (Table 2), establishing a new state-of-the-art among systems using open-weights models.

## 4.3. Critic Model Analysis

In this section, we provide details on important aspects of training critic models, using the performance of 1-step lookahead with 4 action candidates as the benchmark.

Given our sparse reward setting, we use TD($\lambda$) (Sutton & Barto, 1998) to compute training targets. TD($\lambda$) interpolates between pure Monte-Carlo (MC) target estimates and temporal-difference learning, allowing to reduce target variance at the expense of introducing bias into the estimate. Figure 4 illustrates that $\lambda = 0.7$ yields the highest success rate, outperforming both Monte-Carlo estimates ($\lambda = 1$) and one-step TD ($\lambda = 0$). This result suggests that relying exclusively on Monte-Carlo estimates might be suboptimal due to high variance, and a careful search for $\lambda$ might yield benefits in similar setups.

We further analyze how the discount factor $\gamma$ influences the success rate, using a critic trained to predict Monte-Carlo target estimates. As can be seen in Figure 5, as $\gamma$ approaches 1, the success rate increases. Manual trajectory investigation shows that a longer temporal horizon helps the agent avoid missing critical steps (e.g. testing or issue reproduction). Figure 6 corroborates this by demonstrating that higher $\gamma$ values lead to longer trajectories, allocating additional steps to vital yet initially unrewarded actions.

We additionally investigate the critic-estimated action-values for both successful and unsuccessful trajectories, which were produced by the Qwen-based policy with and without 1-step lookahead. Figure 7 illustrates how the aver-

age value estimates evolve as trajectories progress in time, demonstrating critic's ability to differentiate between successful and unsuccessful trajectories. The critic not only assigns higher values to successful trajectories on average but also increases the confidence of its estimates as trajectories progress and additional information becomes available. Naturally, 1-step lookahead causes some amount of value hacking, leading to an increase of average values for negative trajectories. This suggests that building a robust critic model would require multiple iterations of active re-training on adversarial trajectories found by guided search. Finally, there exists a small gap between the positive and negative trajectories at the very beginning, suggesting non-trivial ability to estimate problem complexity from its description.

## 4.4. Co-scaling of $K$ and $T$ for 1-step Lookahead

To better understand the trade-offs between search quality and costs, we investigate how the success rate of 1-step lookahead changes when varying the number of action candidates $K$ and sampling temperature $T$, using a TD(0.8) critic.

As shown in Figure 8, performance initially improves with increasing $K$ across all temperature settings, but saturates at $K = 8$, where our current critic models reach their discriminative ceiling. However, better critic models trained on adversarial trajectories produced by critic-guided search will likely benefit from more than 8 candidates.

The temperature parameter $T$ exhibits different effects at different ranges of $K$. For lower values of $K$, higher temperatures yield better results, indicating the importance of increased exploration when fewer candidates are available. Optimal performance emerges at moderate values of both $K$ and $T$, achieving an effective balance between exploration and exploitation. At high $K$ temperature does not seem to

be very important provided it is not too low and allows for sufficient exploration.

## 5. Conclusion and Future Work

In this work, we introduce the concept of non-serializable RL environments and examine how non-serializability hinders the use of powerful guided search methods. This issue is of practical importance, as Docker containers — commonly used to isolate execution of automated software engineering agents — are one notable example of such environments. Software engineering agents often exhibit a large gap between average-case and best-case performance, and guided search methods can help mitigate this problem.

We describe two guided search techniques, 1-step lookahead and trajectory selection, that operate effectively despite non-serializability constraints. Our empirical study on an automated software engineering benchmark, SWE-bench Verified, reveals that these simple approaches can yield a two-fold performance improvement, achieving state-of-the-art results among systems relying on open-weights models. We further investigate how these methods scale with increased test-time compute and recommend strategies for training critic models to guide them.

Some interesting future directions of this work are:

- Examining whether powerful search methods implemented via replay-based serialization can bring performance benefits despite their lack of correctness.

- Improving the robustness of replay-based serialization, for instance, by disallowing background processes, restricting access to entropy sources to reduce stochasticity, or detecting when replay diverges from the intended state.

- Developing alternative approaches to correctly serialize and deserialize development containers.

## Impact Statement

This work aims to improve the reliability and efficiency of LLM-based agents through guided search strategies. Our method allows to trade additional computation at inference time for higher agent performance, effectively exchanging money for capability. While such techniques allow to achieve better results with existing models using more computation, in the future similar methods can also deepen existing inequalities by giving well-funded actors a further advantage. Next, critic models that we train to guide the agent enable self-verification, thus fostering safer and more reliable agentic systems. Finally, our approach may let open-weight models, when combined with guided search,

match the performance of leading proprietary models, further democratizing AI. We encourage future work to further scrutinize the ethical and distributive impacts of the performance-versus-compute trade-off.

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

## A. Related Work

**Reasoning via Prompting**   Large Language Models have demonstrated strong capabilities in reasoning and planning. Chain of Thought (CoT) (Wei et al., 2022) has emerged as a dominant method for enabling structured reasoning in LLMs by introducing intermediate computation steps. This approach has proven particularly effective for complex tasks, requiring no modifications to model parameters. The effectiveness was further enhanced by the discovery that these behaviors can be elicited in a zero-shot manner through simple prompting techniques (Kojima et al., 2022), demonstrating LLMs' inherent capability for multi-step reasoning. However, single-path generation, even with structured reasoning, may not always yield the best solution.

**Reasoning via Search**   To improve LLM reasoning beyond single-path generation, the simplest approach is to explore multiple solutions without learned evaluation components. Self-consistency (Wang et al., 2023) demonstrated strong performance through sampling multiple complete solutions and implementing Majority Voting over their answers. While effective, this approach relies heavily on random sampling and cannot guide the exploration process.

**Learning to Guide Search**   Combining search with learned value estimation has proven crucial for efficient exploration, pioneered by AlphaGo / AlphaZero (Silver et al., 2016; 2017). This approach has been adapted to LLM reasoning through various search strategies: Beam Search (Setlur et al., 2024), Best-First Search (Koh et al., 2024), Depth-First Search (DFS) (Yao et al., 2023a), and Monte-Carlo Tree Search (Kocsis & Szepesvári, 2006; Putta et al., 2024; Xie et al., 2024; Hao et al., 2023), each offering different trade-offs between exploration breadth and depth.

The development of effective value estimators has been a key challenge in this direction. Early attempts to use LLMs directly as verifiers proved unsuitable for math reasoning (Huang et al., 2023; Luo et al., 2023). Different approaches to value estimation have emerged: some works (Wang et al., 2024a) use Monte-Carlo Tree Search (Kocsis & Szepesvári, 2006), while others (Wang et al., 2024d) employ Monte-Carlo estimation using only policy roll-outs. This data can then be used to train step-level or outcome-level critic models. Outcome-supervised Reward Models (ORMs) (Cobbe et al., 2021; Yu et al., 2024; Mudgal et al., 2024) predict final answer correctness for solution reranking, while Process-supervised Reward Models (PRMs) (Lightman et al., 2024; Wang et al., 2024a; Uesato et al., 2022; Havrilla et al., 2024) evaluate the validity of an intermediate step to rank and select among competing intermediate steps.

**Software Engineering Agents**   Recent approaches to building software engineering agents can be categorized by their reliance on human priors. The first category employs manually defined workflows (Xia et al., 2024; Zhang et al., 2024; Ma et al., 2024; Ehrlich et al., 2025), breaking down complex tasks into specific subtasks and introducing ad hoc verification stages. These approaches require significant engineering effort and lack the ability to adapt to novel situations. The second category builds upon ideas from highly flexible, domain-agnostic frameworks of ReAct (Yao et al., 2023b), which introduced thought-based planning for any multi-step interaction, and CodeAct (Wang et al., 2024b), which proposed using Python code as a universal action interface. SWE-agent (Yang et al., 2024) and OpenHands (Wang et al., 2024c) frameworks adapt these general-purpose ideas to the software development domain, defining an agent-computer interface (ACI) that enables LLM agents to execute arbitrary operations via shell commands.

Although agents in the second category (Schluntz et al., 2024) offer more flexibility, they face challenges in long-term planning. Recent work has shown that combining such frameworks with test-time search strategies can significantly improve performance. For example, SWE-Gym (Pan et al., 2024) studies the benefits of applying outcome supervision to such agents. Our approach, built upon the SWE-agent framework, complements this work by investigating outcome supervision at a significantly larger scale, while additionally studying step-level value estimation and how the two methods can be combined. SWE-Search (Antoniades et al., 2024) leverages MCTS to search the solution space, demonstrating performance scaling with increased test-time computation. This work demonstrates that bypassing stochasticity and non-serializability issues by manually managing a limited subset of the true state (e.g. only filesystem changes in the working copy) can yield meaningful performance benefits. However, we expect this approach to run into issues when applied to scenarios where stochasticity is important, such as dealing with race conditions or interacting with stateful external systems.

## B. Training Details

We train critic models and the base policy using the hyperparameters listed in Table 3.

Critic models are trained with L2 loss, using TD($\lambda$) estimates of discounted reward-to-go as targets. For critic models to

*Table 3.* Hyperparameters used in experiments

| Hyperparameter | Critic | Base policy |
| --- | --- | --- |
| Optimizer | AdamW | AdamW |
| Warmup steps | 7 | 7 |
| Training steps | 459 | 215 |
| Learning rate value | $2 \cdot 10^{-6}$ | $4 \cdot 10^{-6}$ |
| Learning rate schedule | cosine | cosine |
| Batch size | 128 | 128 |
| Number of epochs | 4 | 6 |
| Weight decay | 0.1 | 0.1 |
| Sequence length | 32768 | 32768 |

predict $Q(s, a)$, the unembedding layer of the initial model is replaced with a linear layer that maps token embeddings to scalar outputs. We add a special token to the end of every agent turn, and treat the scalar predicted for this token as the action-value prediction for the turn. Loss is masked for all other tokens during training. All parameters of critic models except the embedding layer are updated during training.

The base policy is trained with conventional cross-entropy loss to maximize the probability of training trajectories. All parameters of the base policy are updated during training, except the embedding and unembedding layers.

## C. GPT-4o-based Policy Results

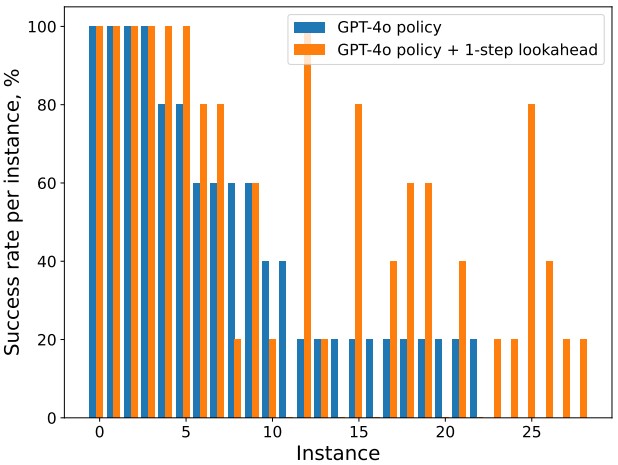

*Figure 9.* 1-step lookahead improves success rate per instance for GPT-4o base policy, adding new solved issues.

*Figure 10.* The dependency between the success rate and the number of candidates $N$ in trajectory selection for GPT-4o.

We analyze the impact of 1-step lookahead on the per-instance success rate for the GPT-4o policy, as shown in Figure 9. Our results indicate that per-instance success rates generally improve across most instances.

Furthermore, we examine the relationship between the success rate and the number of candidates $N$ used for trajectory selection, both for the base policy and when applied on top of 1-step lookahead. The results are presented in Figure 10. As $N$ increases, the success rate continues to improve, with no clear plateau observed within the explored range.

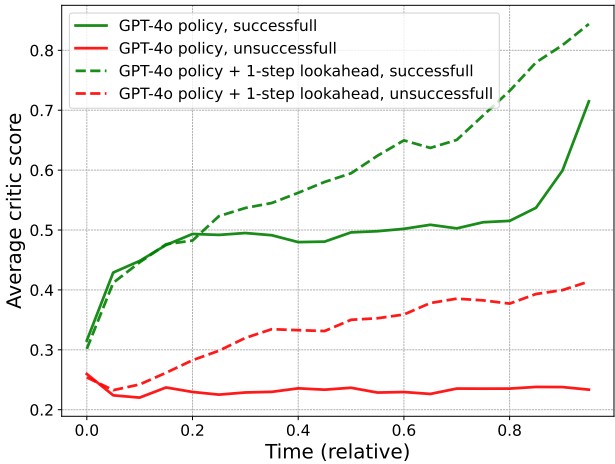

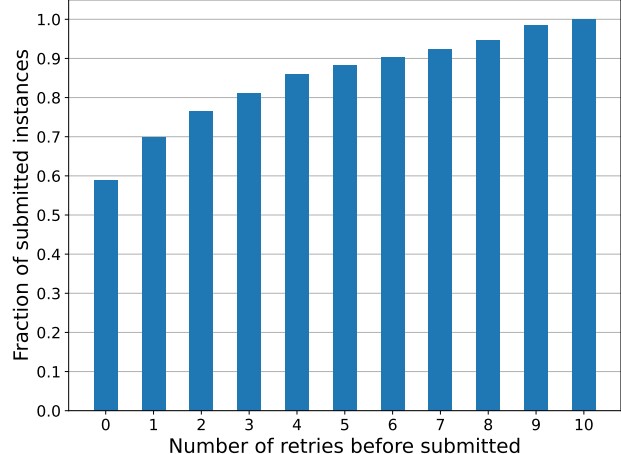

*Figure 11.* Critic learns to distinguish successful and unsuccessful trajectories for GPT-4o.

*Figure 12.* The average number of retries it takes to generate a trajectory that ends with "submit" for a given fraction of the test set using the Qwen-based policy.

Figure 11 illustrates how critic-estimated action values evolve over time for both successful and unsuccessful trajectories generated by the GPT-4o policy, with and without 1-step lookahead. The results suggest that the critic effectively differentiates between successful and unsuccessful trajectories. However, in the case of the GPT-4o base policy, the critic's scores remain stagnant mid-trajectory for successful trajectories, indicating difficulty in recognizing positive steps at intermediate stages. A potential explanation for this behavior is that the critic was not trained on any GPT-4o-generated trajectories, which may have limited its ability to generalize to this setting. Some degree of value hacking is also present with lookahead.

## D. Verified-50 Problems

The Verified-50 dataset, which we curate from the SWE-bench Verified by randomly selecting problems, contains the following problem instances:

```
sympy__sympy-22080, django__django-15315, django__django-11333,
matplotlib__matplotlib-20826, django__django-11532, django__django-16642,
django__django-14855, sphinx-doc__sphinx-8721, pylint-dev__pylint-4604,
sympy__sympy-13615, django__django-13089, django__django-15987,
django__django-14725, sympy__sympy-14248, pytest-dev__pytest-7982,
django__django-15280, scikit-learn__scikit-learn-13142,
pytest-dev__pytest-5809, matplotlib__matplotlib-23299, django__django-16560,
django__django-15103, sympy__sympy-16792, django__django-14007,
psf__requests-2317, django__django-11880, django__django-16136,
django__django-16661, sympy__sympy-17139, sympy__sympy-14531,
sphinx-doc__sphinx-8595, django__django-10880, sympy__sympy-19346,
sphinx-doc__sphinx-9229, django__django-11265, matplotlib__matplotlib-25332,
scikit-learn__scikit-learn-13135, pydata__xarray-6744, pydata__xarray-6461,
sympy__sympy-15017, django__django-13417, matplotlib__matplotlib-24870,
django__django-15368, django__django-11095, django__django-15554,
pydata__xarray-6992, django__django-15863, django__django-13363,
sympy__sympy-13852, django__django-14017, pylint-dev__pylint-4661
```

## E. Running Until Submitted

In our experiments, we employ an iterative execution strategy, in which the agent repeatedly attempts to solve the problem until it issues a "submit" command or reaches a maximum of 10 additional attempts. We refer to this approach, which can

be viewed as a simple form of search, as *until submitted* regime. As shown in Table 4, this strategy significantly improves the performance of the Qwen-based policy, both with and without 1-step lookahead. Additionally, it increases the number of unique successfully solved instances (Pass@$N$), calculated based on the last attempt for each instance, thereby expanding the opportunities for trajectory selection.

Figure 12 shows that the majority of Qwen-based policy solutions are submitted on the first try. However, when 3 additional attempts are allowed, the fraction of submitted solutions increases to $80\%$. This demonstrates that the strategy significantly boosts performance without imposing a substantial computational cost, requiring an average of $\sim 1.65$ extra attempts per problem.

As shown in Table 4, in contrast to the Qwen family, the GPT-4o-based policy is not significantly influenced by the until submitted regime, with the metrics for the default regime almost matching those when running until submitted.

*Table 4.* The effects of running until submitted, Verified-50.

| Inference operator | Default | | | | Until Submitted | | | |
|---|---|---|---|---|---|---|---|---|
| | SR (%) | SEM | Pass@5 (%) | Pass@15 (%) | SR (%) | SEM | Pass@5 (%) | Pass@15 (%) |
| Qwen-based policy | 16.2 | ±1.08 | 33.6 | 44.0 | 22.8 | ±1.05 | 41.8 | 54.0 |
| Qwen-based policy + 1-step lookahead | 26.8 | ±1.08 | 46.4 | 60.0 | 32.4 | ±0.89 | 51.2 | 62.0 |
| GPT-4o policy | 22.0 | ±1.54 | 46.0 | – | 22.0 | ±1.54 | 46.0 | – |
| GPT-4o policy + 1-step lookahead | 27.2 | ±1.20 | 48.0 | – | 29.2 | ±1.62 | 48.0 | – |

## F. Critic Base Model Sensitivity Analysis

*Table 5.* The effects of varying base model for critic on performance of Qwen-based policy + 1-step lookahead, Verified-50, default regime.

| Base model for critic | SR (%) | SEM |
|---|---|---|
| LLaMA3.1-70B | 26.8 | ±1.08 |
| Qwen2.5-72B | 22.8 | ±1.08 |
| LLaMA3.1-8B | 21.0 | ±1.16 |
| Qwen2.5-7B | 20.2 | ±1.08 |
| No critic | 16.2 | ±1.08 |

Table 5 compares critics trained starting from different base models on the task of guiding Qwen-based policy with 1-step lookahead on Verified-50. We don't perform additional hyperparameter tuning for each base model, using hyperparameters selected for LLaMa3.1-70B across all experiments.

Model size appears to be an important factor: smaller models (8B/7B) underperform ($21.0\%$ and $20.2\%$) compared to 70B+ models, hinting at the fact that critic performs non-trivial computations that can benefit from better representations.

In this comparison, LLaMA3.1-70B outperforms its Qwen2.5 counterpart; however, this result may partially reflect unequal hyperparameter tuning. Specifically, we performed extensive hyperparameter optimization for LLaMA3.1-70B, exploring learning rates, batch sizes, and training schedules. In contrast, for Qwen2.5-72B, we applied the best configuration identified

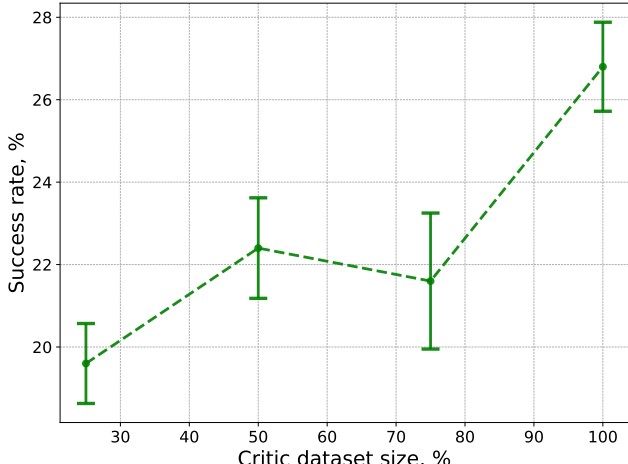

*Figure 13.* The effects of varying training dataset size for critic on performance of Qwen-based policy + 1-step lookahead, Verified-50, default regime.

for LLaMA3.1-70B. We recommend targeted hyperparameter tuning for the specific model family used to maximize performance of the trained critic model.

## G. Sensitivity Analysis for Critic Dataset Size

Figure 13 presents a sensitivity analysis of critic performance with respect to training dataset size, with critic evaluated on SWE-bench Verified-50 using 1-step lookahead in default regime. We systematically reduce the training set size from $100\%$ to $25\%$ to understand how varying data sizes affect critic quality. We don't perform additional hyperparameter tuning for each dataset size, using hyperparameters optimized for the full dataset across all experiments.

As expected, critic performance tends to decline with dataset size, with the highest success rate of $26.8\% \pm 1.08$ observed when using the full dataset. The performance drop on smaller subsets at least partially comes from insufficient training signal. However one other reason likely contributing to performance drop is that on smaller subsets, bootstrapped action-value estimates used in TD($\lambda$) targets simply don't have enough time to stabilize. We expect that by explicitly tuning hyperparameters for smaller datasets one can achieve better critic performance.

