# OpenReview forum: "Guided Search Strategies in Non-Serializable Environments with Applications to Software Engineering Agents"
_ICML.cc/2025/Conference — ICML 2025 poster_

### Official Review · Reviewer_gMAZ · 2025-03-13

**Overall Recommendation:** 4

**Summary:**

This paper investigates guided search strategies for Large Language Models (LLMs) in non-serializable environments, focusing on SWE agents. The authors identify a significant gap between the average-case and best-case performance of LLMs in complex multi-step tasks. While effective search techniques like Monte Carlo Tree Search (MCTS) exist, they cannot be directly applied in non-serializable environments (like Docker containers) where intermediate states cannot be saved and restored.

The paper proposes two complementary search strategies applicable to such environments:
1. 1-step lookahead: Guided by a learned action-value function to select the most promising next action
2. Trajectory selection: Selecting the most promising complete trajectory from multiple attempts

When applied to the SWE-bench Verified, these methods doubled the average success rate of a Qwen-72B-based model, achieving 40.8% success - a new state-of-the-art for open-weights models. The techniques also showed similar improvements when applied to GPT-4o.

**Claims And Evidence:**

The main claims are well-supported by:

- Theoretical framework describing non-serializability constraints and how they limit search strategies
- Comprehensive experiments showing performance improvements on SWE-bench Verified
- Ablation studies examining different parameters (λ values for TD learning, discount factor γ, number of candidates K, temperature T)
- Analysis of how their methods improve per-instance success rates
- Comparison with state-of-the-art results for open-weights models

The paper provides substantial empirical evidence showing that their proposed search strategies significantly improve performance both individually and when combined.

My only concern about these evidence was related to an experiment setting- I will elaborate on this in a later section.

**Essential References Not Discussed:**

N/A

**Experimental Designs Or Analyses:**

This is my only major concern on the result:

> However, we observe that context exhaustion frequently correlates with unrecoverable mistakes mid-trajectory, leading to incorrect solutions. To enhance reliability, we implement an iterative execution strategy: the agent runs repeatedly until it terminates by issuing a “submit” command or reaches a maximum of 10 additional attempts.

I like this paper, but I think this is not a fair setting for comparison: I would consider this as best-of-K strategy (where K is the number of re-runs) with a rule-based verifier that selects trajectories with a "submit" action.

I would recommend the authors to also present the main experimental results in Tables 1 and 2 *without* this setting for the reader to have a clear understanding of whether it impacts the conclusion of this paper.

I'd be happy to raise the score if we can observe the same performance trend **without doing until submit**

**Methods And Evaluation Criteria:**

The methods are sensible for the problem at hand:

- Well-defined non-serializable environment framework
- Clear implementation of 1-step lookahead and trajectory selection algorithms
- Appropriate training of critic models using TD(λ) learning with extensive ablation that help the reader understand the importance of each components
- Use of SWE-bench Verified for evaluation, which aligns with the software engineering focus

**Other Comments Or Suggestions:**

- For figure 1, I would consider that number "pass@N" as opposed to "best@N"

**Other Strengths And Weaknesses:**

N/A

**Questions For Authors:**

N/A

**Relation To Broader Scientific Literature:**

This paper builds upon prior work in guided search for LLMs / inference-time scaling while addressing the specific challenges of non-serializable environments (e.g., specifically in software engineering).

**Theoretical Claims:**

N/A

---

> ### Author Rebuttal · Authors · 2025-03-31
>
> We thank the reviewer for thoughtful comments and valuable feedback.
>
> We agree that Pass@N is the correct term for trajectory selection with oracle and will change the text and plots accordingly.
>
> We respectfully disagree that “run until submitted” is not a fair setting for evaluation: all evaluations done in the paper have been done using this setting unless stated otherwise. The results without running until submitted have been included in the Appendix E, which shows that lookahead search still has a dramatic effect on agent performance even when not using this regime. However we agree that the presentation of results without running until submitted was incomplete and will add trajectory selection results to this section. For your reference, turning off “running until submitted“ has the following effect on trajectory selection performance:
>
> * Qwen-based policy: 22.8% → 18.1%
> * Qwen-based policy + trajectory selection (N = 5): 32.3% → 27.2%
> * Qwen-based policy + trajectory selection (N = 10): 37.6% → 31.3%
> * Qwen-based policy + 1-step lookahead + trajectory selection (N = 5): 40.0% → 36.5%
> * Qwen-based policy + 1-step lookahead + trajectory selection (N = 10): 44.1% → 41.7%
>
> We also agree that a subset of these results should be added to table 1 for clarity. We are not certain that these results should be added to table 2, which goal is to compare our best setup to concurrent work.

---

### Official Review · Reviewer_RSYG · 2025-03-14

**Overall Recommendation:** 4

**Summary:**

The paper introduces and formalizes the notion of a non-serializable environment. Traditional guided search techniques cannot be applied to non-serializable environments. A particular instance of a non-serializable environment is mutable software engineering environments, where it can be difficult to revisit prior states or roll back changes.

The authors study two guided search methods that are applicable to non-serializable environments (SWE-Bench-Verified). The methods are one-step-lookahead and trajectory selection using a learned critic. To train a critic, the authors construct an auxillary dataset similar to SWE-Bench and use it to generate trajectories for training a critic. Combined with one-step lookahead, the critic improves policies in the SWE-Agent scaffolding by a substantial amount.

**Claims And Evidence:**

Yes.

**Essential References Not Discussed:**

None.

**Experimental Designs Or Analyses:**

I checked all experiments.

**Methods And Evaluation Criteria:**

Yes.

**Other Comments Or Suggestions:**

None.

**Other Strengths And Weaknesses:**

I think the only weakness of the paper worth mentioning is that the paper only shows results for a single open model (Qwen-72B-Instruct).

The strength of the paper is that it addresses an important problem (non-serializable environments) by applying simple, effective methods from the RL literature. The method works, and the authors collected their own dataset to train a critic.

**Questions For Authors:**

1. Why not also try this with a SLM (7B parameters)? You could take the offline dataset you generated with the 72B model or GPT-4o and train on it.
2. Why do you use a different LLM family for the critic?

**Relation To Broader Scientific Literature:**

This paper is most closely related to LLM-driven search [1], process supervision for reasoning [2] and training software engineering agents [3]. LLM-driven search methods typically rely on serializable environments, and this paper studies non-serializable environments. Thus far, process supervision for LLM reason has mostly been applied only to mathematics task (and possibly simple coding tasks). Training software engineering agents is relatively new, and I'm not aware of any works that are not concurrent work that apply process supervision for training software engineering agents.


[1] Hao, Shibo, et al. "Reasoning with language model is planning with world model." arXiv preprint arXiv:2305.14992 (2023).
[2] Lightman, Hunter, et al. "Let's verify step by step." The Twelfth International Conference on Learning Representations. 2023.
[3] Pan, Jiayi, et al. "Training Software Engineering Agents and Verifiers with SWE-Gym." arXiv preprint arXiv:2412.21139 (2024).

**Theoretical Claims:**

Not applicable.

---

> ### Author Rebuttal · Authors · 2025-03-31
>
> We thank the reviewer for thoughtful comments and valuable feedback.
>
> Regarding the question about the reasons for choosing LLaMa3 70B as the base model for training critic, they are purely historical: this was the best 70B model available at the time when this project started, which gave a reasonable balance of model size and capability. We agree that this might be confusing and the paper can benefit from an ablation on model sizes and families. We will train critic models using the same data mixture on 2 model families (LLaMa 3 and Qwen 2.5) and 2 model sizes (7B and 70B) and present results in the appendix. The corresponding models are already in the process of being trained.

---

### Official Review · Reviewer_uvAg · 2025-03-17

**Overall Recommendation:** 3

**Summary:**

This paper proposes training a critic for action-value estimates for guided search in non-serializable environments (SWEbench). The high variance of LLM agents performance on this benchmark is well known, and the proposed trained  critic can address this issue while enabling inference-time scaling through guiding the search process, in 1) 1-step lookahead, 2) best-of-n trajectory selection and 3) combination of both.

**Claims And Evidence:**

The experiment results on SWEbench show that the trained critic enables doubling of performance on SWEbench verified. Analysis experiments show that 1step lookahead reduces the performance variance, and analysis of the critic model shows that it exhibits several positive properties, including scaling to more number of action candidates, better distinguishes successful and unsuccessful trajectories, and is able to reflect long-horizon value.

**Essential References Not Discussed:**

A missing reference on SWE agents with guided search is SWE-Search: Enhancing Software Agents with Monte Carlo Tree Search and Iterative Refinement, ICLR 2025

**Experimental Designs Or Analyses:**

i did not identify any serious issues.

**Methods And Evaluation Criteria:**

yes

**Other Comments Or Suggestions:**

- If the authors can include results for smaller models, it would demonstrate the generalizability of the proposed approach.
- Did you try using an off-the shelf prompted LLM to do 1-step lookahead or best-of-n trajectory selection? If feasible, it would be interesting to see how this compares with the training approach.

**Other Strengths And Weaknesses:**

The paper shows strong results in a challenging domain. While verifier-based inference time scaling is an existing idea, there are few which successfully apply it to SWEbench. The paper is clearly written and easy to follow.

A weakness of the paper is lack of details on training and cost details:
- It's confusing whether the policy is finetuned or not: In table 1, is Qwen-based policy + 1-step lookahead a finetuned policy + 1-step lookahead?
- It's unclear whether training is full-parameter or not, with cost implications as the finetuned model is on the larger side with 72B

**Questions For Authors:**

- What is the "LLM-based policy trained in-house"? Is the base policy e Qwen-72B-Instruct or something else?
- How important is the 80k number for the critic training data? Did you try smaller or larger training sets?
- Do you do full-paramemter finetuning?

**Relation To Broader Scientific Literature:**

the paper is related to SWE agents and inference-time scaling.

**Theoretical Claims:**

n/a

---

> ### Author Rebuttal · Authors · 2025-03-31
>
> We thank the reviewer for thoughtful comments and valuable feedback.
>
> We thank the reviewer for pointing out the missing reference to “SWE-Search: Enhancing Software Agents with Monte Carlo Tree Search and Iterative Refinement, ICLR 2025”. We will add the reference to the related work section and discuss its connection to the present work.
>
> We explain that “Qwen-based policy“ is a finetuned policy based on Qwen-2.5-72B-Instruct in section 4.1, paragraph “training details“. Experimenting on an open model is a necessity for this project to reduce inference costs, however experimenting on a non-finetuned instruct model is not an option: SWE-bench is a hard benchmark and current generation instruct models don’t perform too well on it out-of the box, hindering our ability to evaluate the effect of interventions we make. We agree that this aspect of the present work can benefit from a more clear presentation and will try to improve it in the final version of the paper.
>
> All training we do is full-parameter, but it was indeed not clarified in the paper. We will add this detail to the training setup description. We believe that comparing LoRA vs full finetuning for training critics lies outside of the scope of this paper, as our method is orthogonal to fine-tuning methodology.
>
> We agree that this work can benefit from an ablation on model sizes. We will train critic models using the same data mixture on 2 model sizes (7B and 70B) and present results in the appendix. The corresponding models are already in the process of being trained.
>
> We also agree that the paper can benefit from an ablation of data mixture size. We will train critic models on 25%, 50% and 75% of the data mixture we used and will report the performance of lookahead and trajectory selection methods in the appendix. The corresponding models are already in the process of being trained.
>
> Regarding using off-the-shelf prompted LLM as a critic for lookahead/trajectory selection, we did experiment with such approaches but couldn’t produce a critic with performance significantly exceeding random guessing. We decided not to include these results as a baseline because it wasn’t clear if this was a skill issue on our side.

---

### Official Review · Reviewer_LbVW · 2025-03-24

**Overall Recommendation:** 3

**Summary:**

This paper introduces two algorithms for guiding LLM policies in non-serializable RL environments using a learned action-value estimator (LLaMA-3-70B) as a critic. The proposed methods are: ***1-step lookahead**, which acts as a process-level reward model by selecting actions at each step, and **Trajectory Selection**, which serves as an outcome-level reward model by evaluating entire trajectories. The authors demonstrate that their test-time guided search methods can improve performance on SWE-bench Verified tasks of LLM policies (Qwen-72B-finetuned, GPT-4o).

**Claims And Evidence:**

The claims made in the submission are supported by clear and convincing empirical evidence.

**Essential References Not Discussed:**

None that I identified. The paper appears to situate itself well within the relevant literature.

**Experimental Designs Or Analyses:**

The experimental design is rigorous and clearly presented. I found no methodological issues with the evaluation. In particular, the ablations and comparisons (e.g., best-of-N) are valuable for contextualizing the utility of the proposed methods.

**Methods And Evaluation Criteria:**

The proposed methods and evaluation setup are appropriate for the problem of guided search in non-serializable environments.

**Other Comments Or Suggestions:**

No further suggestions or minor issues to report.

**Other Strengths And Weaknesses:**

Strengths:
- The writing is clear and accessible.
- Algorithm design is intuitive and well-motivated.
- Empirical results are presented with care and depth.

Weaknesses:
- The reliance on training a critic using 80k labelled trajectories raises practical concerns regarding data collection and compute costs. It would be helpful to better contextualize the cost-benefit trade-off between this approach and a simpler best-of-N sampling strategy, especially given that best-of-3 GPT-4o performs comparably to the more complex guided search method in some cases (e.g., Figure 10).

**Questions For Authors:**

1. What was the rationale behind choosing LLaMA-3-70B as the critic model?
It would be useful to understand whether this choice was based on empirical performance (e.g., ablation over model sizes), availability, or alignment properties. Clarification here would help assess whether a smaller or less resource-intensive model could have been similarly effective.

**Relation To Broader Scientific Literature:**

The paper makes a contribution by applying action-value-based test-time guidance to non-serializable RL environments, a relatively underexplored area. This complements recent work on Software Engineering Agents and extends ideas from traditional RL into new LLM application domains.

**Theoretical Claims:**

No formal proofs are provided, but the presented formulas align well with the textual descriptions and intended algorithmic behavior. I found the derivations sound and consistent with expected outcomes.

---

> ### Author Rebuttal · Authors · 2025-03-31
>
> We thank the reviewer for thoughtful comments and valuable feedback.
>
> Regarding the comment about the trade-off between the lookahead and the trajectory selection methods (assuming that is what the reviewer means by “best-of-N sampling strategy”), we’d like to emphasize that trajectory selection still requires a trained critic. Admittedly, this critic is not required to have process supervision abilities, which in theory can make its training more sample efficient. Therefore, we agree with the reviewer (and other reviewers who have also emphasized this point) that the paper can benefit from an ablation of data mixture size. We will train critic models on 25%, 50% and 75% of the data mixture we used and will report the performance of both lookahead and trajectory selection methods in the appendix. The corresponding models are already in the process of being trained.
>
> Regarding the question about the reasons for choosing LLaMa3 70B as the base model for training critic, they are purely historical: this was the best 70B model available at the time when this project started, which gave a reasonable balance of model size and capability. We agree that this might be confusing and the paper can benefit from an ablation on model sizes and families. We will train critic models using the same data mixture on 2 model families (LLaMa 3 and Qwen 2.5) and 2 model sizes (approx 7B and 70B) and present results in the appendix. The corresponding models are already in the process of being trained.

---

### Decision · Program_Chairs · 2025-05-01

**Decision:**

Accept (poster)

**Comment:**

The paper tackles the problem of guiding LLMs in non-serialisable environments (i.e. environments where states cannot be saved, reloaded or rewound), which makes search methods like MCTS inapplicable. The key motivation here is software engineering tasks, which are run in Docker containers, where states simply cannot be rolled back.  To address this type of domain, the authors propose two guided search strategies for LLM agents: one-step-lookahead and trajectory selection using a learned critic. To train a critic, an auxiliary dataset similar to SWE-Bench is used to generate trajectories for training. These methods significantly improve performance on SWE-bench, doubling the performance of several LLMs on the benchmark.

Reviewers all agree that the paper is well-written, thoughtful and shows impressive results on a very hard benchmark task, and so would make a strong contribution to the community. The paper could benefit from more ablation studies, which were identified during the rebuttal phase and which the authors are in the process of running. These results could further strengthen what is already a solid paper, and the authors are encouraged to include them in a finalised version.